



# Evaluation of coastal Antarctic precipitation in MAR3.9 regional and LMDz6 global atmospheric model with ground-based radar observations

Florentin Lemonnier[a,1], Alizée Chemison[b], Hubert Gallée[c], Gerhard Krinner[c], Jean-Baptiste Madeleine[a], Chantal Claud[a], and Christophe Genthon[a]

[a]Sorbonne Université, École normale supérieure, PSL Research University, École polytechnique, CNRS, Laboratoire de Météorologie dynamique, LMD/IPSL, F-75005 Paris, France
[b]Laboratoire des Sciences du Climat et de l'Environnement, CNRS-CEA-UVSQ – UMR8212, CE Saclay, France
[c]Université Grenoble Alpes, CNRS, Institut des Géosciences de l'Environnement, Grenoble, France

**Correspondence:** Florentin Lemonnier (flemonnier@lmd.jussieu.fr)

**Abstract.**

In the current context of climate change in the poles, one of the objectives of the APRES3 (Antarctic Precipitation Remote Sensing from Surface and Space) project is to characterize the vertical structure of precipitation in order to better simulate it. Nowadays, the precipitation simulated by models in Antarctica is very widespread and overestimated the data. Sensitivity

5    studies have been conducted using two models and compared to the observations obtained at the Dumont d'Urville coast station, obtained by a Micro Rain Radar (MRR). The MAR meso-scale model specifically developed for the polar regions and the LMDz/IPSL general circulation model, with zoomed configuration over Dumont d'Urville, have been considered for this study. These models being different in resolution and physical configuration, performing an inter-comparison required numerical, dynamic and physical adjustments in LMDz. A sensitivity study was conducted on the physical and numerical parameters of

10   the LMDz model and on the resolution of the MAR with the aim of estimating their contribution to the precipitation simulation. Sensitivity tests with MAR revealed that this model is well adjusted for precipitation modeling in polar climates, this confirming that this model is a reference in polar climate modeling. Regarding LMDz, sensitivity experiments revealed that modifications in the sedimentation and sublimation parameters do not significantly impact precipitation rate. However, dissipation of the LMDz model, which is a numerical process that dissipates spatially excessive energy and keeps the model stable, impacts





precipitation indirectly but very strongly. A suitable adjustment of the dissipation reduces significantly precipitation over Antarctic peripheral area, thus providing a simulated profile in better agreement with the MRR observations.

*Copyright statement.* Author(s) 2019

## 1 Introduction

Between 1880 and 2012, the Earth's mean global temperature increased by 0.85±0.2°C, and this warming is predicted to intensify during the $21^{st}$ century. As temperatures warm, sea level rises as continental ice melts and the oceans expand thermally. Sea levels have already increased by 190±20 mm between 1901 and 2010 and the Antarctic contribution is estimated at 0.27 mm.yr$^{-1}$ (Church et al., 2013). Antarctica has already lost 2720±1390 billion tonnes of ice between 1992 and 2017 (Shepherd et al., 2018). To understand the impact of the Antarctic ice cap on mean sea level, it is essential to calculate its mass balance.

Precipitation represents the only positive contribution of the surface mass balance, but is difficult to assess over this continent. Precipitation estimates are inferred from surface accumulation observations during field campaigns, but is inhibited by high wind speeds over the ice-sheet leading to under-estimation of the snow accumulation (Das et al., 2013). It is also observed from space with the CloudSat satellite (Palerme et al., 2014) and recent studies have greatly improved confidence in the results of this satellite (Souverijns et al., 2018; Lemonnier et al., 2019). However, the observations are unavailable below 1200 meters above

the surface due to contamination of radar reflections by icy surfaces (Palerme et al., 2019). There are also in-situ observations of precipitation measurements and snow accumulation. However, field campaigns allowing this are difficult to be conducted and are mainly located near the coast (Eisen et al., 2008).

Climate models allow to analyze and understand dynamical and physical processes, such as precipitation, and then to predict the future climate of Antarctica. Different types of climate models exist, ranging from basic 1D models to meso-scale and

coupled global climate models. These models provide a better understanding of the current climate with its fluctuations, as well as a prediction of future climate change. This ability to predict climate change makes it a particularly interesting tool, notably for the Coupled Model Intercomparison Project (CMIP, Taylor et al. (2012)) in the current situation of global warming. These models have different uses, depending on whether they are global or regional, as well as different levels of complexity and various horizontal and vertical resolutions. The calculation time is crucial, so a regional model can easily include developed

and complex physical processes, while a global model has to provide suitable simulations in any region of the globe thus limiting the complexity of the processes it integrates.

Most climate models predict that the Antarctic ice sheet surface mass balance is subject to increase due to higher precipitation rate, which is itself associated with an increase in atmospheric temperature (Krinner et al., 2008). This change in precipitation ranges from 5.5 to 24.4 % during the $21^{st}$ century, depending on greenhouse gas emissions exercises. However, the Palerme

et al. (2017) study presenting an intercomparison of CMIP5 models with CloudSat observations and ERA-Interim reanalysis shows that the models overestimate precipitation in comparison with CloudSat climatology (Palerme et al., 2014), sometimes





by more than 100%. And even though the simulated surface precipitation is compared to an observation level at an altitude of 1200 meters above the local surface, the discrepancy between data and models is large, and questionable for the future prediction of precipitation. In addition, the agreement between data and models is even worse for the simulation of precipitation on the plateau than over the peripheral regions (Palerme et al., 2017; Roussel et al., 2019).

Since November 2015, during a field campaign at the French base in Dumont d'Urville, instruments have been installed, including a Micro Rain Radar (MRR) observing clouds and precipitation particles from surface (Grazioli et al., 2017a). This instrument has provided a continuous vertical structure of precipitation and its climatology. Among other results, this has highlighted the sublimation of precipitation by katabatic winds, as well as providing information on the mean sedimentation rate of precipitation (Grazioli et al., 2017b; Durán-Alarcón et al., 2019). This vertical profile is also an excellent tool for
evaluating the simulated vertical structure of precipitation.

   In this study, we propose to evaluate the vertical structure of precipitation at Dumont d'Urville, simulated by two different models with the MRR dataset. The first model is the general circulation model LMDz, an atmospheric component of the coupled IPSL model. The second model is the mesoscale model Modèle Atmosphérique Régional (MAR). Each of these models having different degrees of complexity because of different uses, it is important to verify how precipitation is simulated by these
two models, and especially to verify if the vertical profile of precipitation is in agreement with the observed profile. In section 2, each model configuration and the ground radar observations are presented to do this study. The sensitivity experiments performed on each model and their results are discussed in section 3. Then, an exploration of numerical dissipation in the LMDz model applied to temperature and its impact in simulated precipitation is discussed in section 4. Finally, we conclude this study in section 5.

## 2   Methods

### 2.1   The LMDz-IPSL climate model

The LMDz dynamical core is based on finite difference and finite volume discretization of the primitive equations of meteorology and transport equations, coupled to a set of physical parameterizations (Hourdin et al., 2013). The radiative transfer scheme is the Rapid Radiative Transfer Model (RRTM) from Mlawer et al. (1997), also used in MAR. The microphysical cloud
scheme is from statistical type and includes large scale condensation. A fraction $f_{iw}$ of the condensed water $q_c$ is assumed to be frozen, depending on the temperature between 273.15 K where $f_{iw} = 0$ and 243.15 K where $f_{iw} = 1$. Then a fraction of the condensed water is partially precipitated according to Zender and Kiehl (1997). The associated sink of cloud water is:

$$\frac{dq_{iw}}{dt} = \frac{1}{\rho}\frac{\partial}{\partial z}\left(\rho w_{iw} q_{iw}\right) \tag{1}$$

where $w_{iw} = \gamma_{iw} \times w_0$, $w_0 = 3.29(\rho q_{iw})^{0.16}$ being the characteristic sedimentation rate of ice crystals given by Heymsfield
and Donner (1990) depending on the solid cloud water and $\gamma_{iw}$ being a tunable parameter. Precipitation is then re-evaporated





and included into the vapor water following :

$$\frac{\partial P}{\partial z} = \beta \left( 1 - \frac{q}{q_{sat}} \right) \sqrt{P} \tag{2}$$

where $P$ is the precipitation flux and $\beta$ is a tunable parameter.

This model configuration only admits the atmospheric model, without taking into account vegetation or ocean circulation
models. However, there is a surface scheme. It is composed of four categories: oceans, continental surfaces, sea-ice and glaciers.
The surface fluxes are calculated by taking into account the parameters of each type of surface. It is important to note that for
desert surfaces such as ice caps, a skin effect model is used to describe surface flows. In order to have the better resolution
possible above Dumont d'Urville with a GCM, the model is stretched longitudinally and latitudinally, reaching a horizontal
resolution of ∼25 km. We nudged the LMDz model with wind, temperature and humidity ERA-Interim reanalysis outside the
zoom. It is nudge-free inside the zoomed area (Coindreau et al., 2007). This makes it possible to represent the processes that
are part of the LMDz model in atmospheric situations that are close to the real condition of the atmosphere. It has 79 vertical
levels in its current configuration, with refinement in the boundary layer and troposphere. The vertical precipitation profile
studied at Dumont d'Urville in the LMDz model is selected over continental surface. A spin-up of 4 months is necessary to
balance the model, then each simulation is conducted for one month corresponding to our dataset period.

**2.2  The Modèle Atmosphérique Régional (MAR)**

MAR is a primitive equations hydrostatic model, developed for polar regions studies. Its dynamical core (Gallée and Schayes,
1994) and its turbulent scheme (Duynkerke, 1988) are designed to replicate classical linear mountain waves such as katabatic
winds. The hydrological cycle includes a cloud microphysical model, with conservation equations for cloud droplet, rainfall,
cloud ice crystal and snow flake distributions. Blowing snow is included in this scheme and these particles are considered as
snow flakes. The sublimation of snow flakes is function of the ice relative humidity, according to (Lin et al., 1983). The MAR
model represents accurately the atmospheric boundary layer, blown snow processes and their interactions with katabatic winds
(Agosta et al., 2019). However, we have not enabled this in our study to save computing time because the processes of interest
occur at higher altitudes. The representation of the cloud microphysical processes is essentially based on the parameterizations
of Kessler (1969). The atmospheric component of MAR is coupled to a snow pack model (Gallée and Duynkerke, 1997),
enhanced by metamorphism laws of the CROCUS snow model (Brun et al., 1992).

MAR is forced in wind, temperature and humidity with ERA-Interim reanalysis outside the domain of study. The optimal
configuration of MAR is a 5 km horizontal resolution with a considered domain of 1000x1000 km, as well as 40 vertical levels
between the surface and the top of the troposphere (about 8000 m above sea level). MAR is accurate on the surface and in the
boundary layer, the first level being 15 cm from the surface. The vertical precipitation profile studied at Dumont d'Urville in
MAR is selected over continental surface. The MAR model requires a shorter spin-up than the LMDz, however to facilitate
comparison with the LMDz, the same duration is chosen.



### 2.3 Micro Rain Radar (MRR) observations

The MRR is a vertically profiling Doppler radar operating at a frequency of 24.3 GHz (K-band) and having a beamwidth of 2° (around 50 m in diameter at 3000 m). The vertical resolution is set to 100 m per bin ranging from 300 – first valid available measurements – to 3000 m (Grazioli et al., 2017a). The MRR's raw measurement – Doppler spectral densities – are available at 10s temporal resolution then minute averaged. The collected data are processed using the IMProTool developed by Maahn and Kollias (2012). The radar reflectivity derived from MRR was calibrated by comparison with a colocated X-band polarimetric radar over the period from December 2015 to January 2016 (for more details, see Grazioli et al. (2017a)). Through this calibration with the second radar, the reflectivity (at X-band) is converted into snowfall rates using a radar reflectivity $Ze$ / snowfall rate $Sr$ relation (Grazioli et al., 2017a) :

$$Z_e = 76 * S_r^{0.91} \tag{3}$$

with $Z_e$ the radar reflectivity (in dBZ) and $S_r$ the snowfall rate (in mm/hr). Grazioli et al. (2017a), proposed a range of values of [69-83] for the prefactor and [0.78-1.09] for the exponent corresponding to a confidence interval of 95 %.

The period selected for this study is February 2017. During this period precipitation events are particularly frequent with different amplitudes and durations. Rather than studying a particular event, we focus on the monthly accumulation of precipitation at each vertical level of the MRR. The monthly accumulation of precipitation is presented in figure 1. The sublimated part of the precipitation can be clearly observed below 1000 meters, due to katabatic winds (Grazioli et al., 2017b).

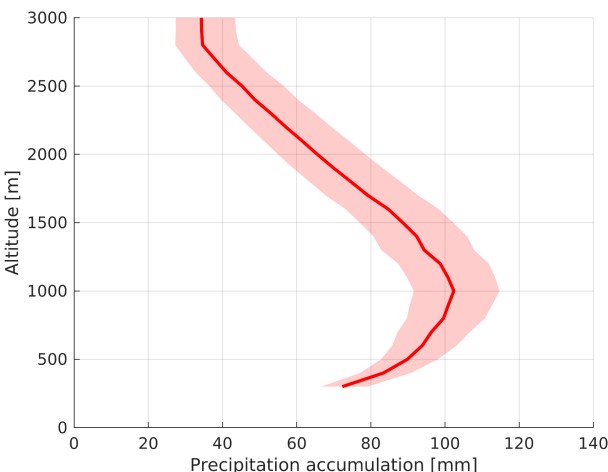

**Figure 1.** Vertical precipitation accumulation over the February 2017 period recorded by the MRR. Red filled area corresponds to the 95% confidence interval of the MRR observations.

a)                                                        b)

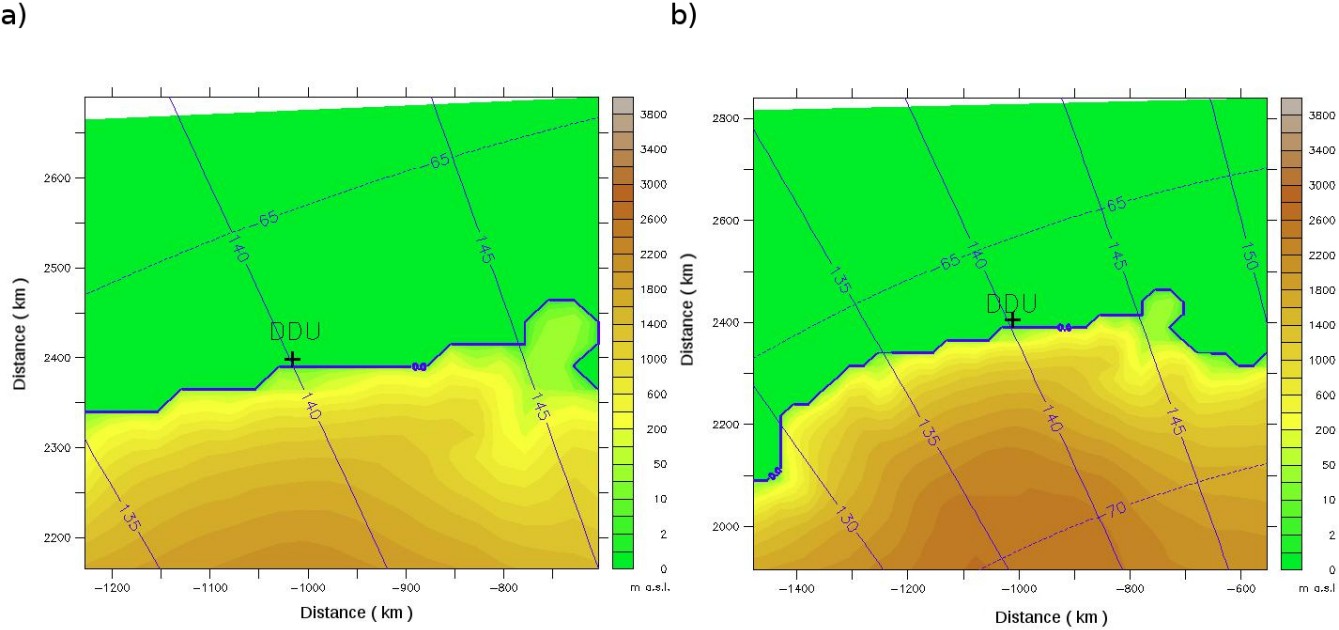

**Figure 2. a)** Topographical representation of the MAR *SMALL* domain. **b)** Topographical representation of the MAR *BIG* domain.

## 2.4    Description of sensitivity experiments

Mesoscale models are very sensitive to horizontal resolution, as the consideration of many parameterizations will strongly depend on it. We used different dimensions of the grids as well as the size of the domain under study. Two domains are presented: the first is 250 km x 250 km, the second is 1000 km x 1000 km, and they are called respectively *SMALL* and *BIG*.

These areas are presented in figure 2. Two horizontal resolutions are also used. The first one is 5 km and the second is a coarser resolution of 25 km. Three simulations have been conducted. The first one, which corresponds to the standard simulation, has a *BIG* domain and a fine resolution of 5 km, the second has a *BIG* domain and a resolution of 25 km and the third has a *SMALL* domain and a resolution of 25 km. The second simulation aims at differentiating the impact of the horizontal resolution of the domain size. The purpose of the two last simulations is to get as close as possible to the LMDz horizontal configurations in

order to compare both models. Although the last two of these configurations are difficult to achieve in previous versions of the MAR model (Franco et al., 2012), the latest version of this model allows these simulations to be run as "critical" cases of MAR use.

As for MAR on figure 2, we evaluated the horizontal resolution of LMDz by performing simulations on two zoomed domains of different sizes. Indeed, when zooming with the LMDz model, the zoomed region can be widened, so we were able to

reproduce the two domains evaluated for MAR. The size of the "*SMALL*" zoom domain in LMDz allows the model to adapt its own physics inside the zoom in an environment where large-scale wind, temperature and humidity advections are controlled by





ERA-Interim reanalyses. The second configuration with a *BIG* domain is larger, so the model can have its own mesocyclonic circulations within the zoom. The center of the zoom is in this case not very affected by the ERA-Interim reanalysis.

The first sensitivity experiment is evaluating the feedback of the LMDz model to the extent of the nudged-free zoomed domain. Indeed, in the case where the zoom area is restricted in size, the center of the zoom is very sensitive to forcing outside this area. This case is similar to a regional climate model. Inversely, when the zoom area is large, the center of the zoom area is less affected by the forcings imposed on it from the outside and the model is more like a global climate model in a free configuration.

The second experiment studies the sensitivity of solid precipitation to sedimentation velocity rate. To do so, we have tested different values of the parameter $w_{iw}$ in the equation 1 through its parameter $\gamma_{iw}$. The different imposed values are summarized in table 1. It is important to note the difference between experiment *SedEx 02* whose sedimentation rate tends towards $1$ m.s$^{-1}$ and the experiment *SedEx 03* whose sedimentation rate is equal to $1$ m.s$^{-1}$ (see equation 1). Indeed, the value of $w_0$ is varying with $q_{iw}$ and the air density as a function of pressure and temperature. In the *SedEx 03* experiment, this variation is not taken into account.

**Table 1.** Sedimentation rate experiments on LMDz precipitation simulation.

| Experiment | Sedimentation rate |
| --- | --- |
| Control simulation | $\gamma_{iw} w_0 \rightarrow 0.25$ m.s$^{-1}$ |
| SedEx 01 | $\gamma_{iw} w_0 \rightarrow 0.5$ m.s$^{-1}$ |
| SedEx 02 | $\gamma_{iw} w_0 \rightarrow 1$ m.s$^{-1}$ |
| SedEx 03 | $\gamma_{iw} w_0 = 1$ m.s$^{-1}$ |

The third sensitivity study with LMDz has been performed on the precipitation sublimation equation 2. To do this, several orders of magnitude have been fixed to $\beta$ tunable parameter value. These values are summarized in the table 2.

**Table 2.** Sublimation tunable parameter experiments on LMDz precipitation evaporation.

| Experiment | $\beta$ sublimation parameter |
| --- | --- |
| Control simulation | $\beta = 2.10^{-4}$ |
| SubEx 01 | $\beta = 4.10^{-4}$ |
| SubEx 02 | $\beta = 8.10^{-4}$ |
| SubEx 03 | $\beta = 2.10^{-3}$ |



## 3 Results of sensitivity experiments

### 3.1 Horizontal resolution and domain size in MAR

We examined the sensitivity of MAR with two different horizontal resolutions and two different domain sizes. The studied horizontal resolutions are the 5 km standard resolution of MAR and a 25 km resolution analogous to LMDz. The vertical
precipitation accumulation profiles simulated by MAR and compared to the MRR are presented in figure 3.

Green dashed line corresponds to best MAR configuration with a 5 km horizontal resolution and a *BIG* domain is in good agreement with MRR vertical observed profile. In this configuration, the model reproduces very well the sublimated part of the precipitation due to the katabatic winds, as already studied by Gallée and Pettré (1998). The inversion point is located at the same altitude as the observed profile. However, too much precipitation is simulated at high altitude, this being a characteristic
bias of the models, highlighted by Grazioli et al. (2017b).

Green solid line corresponds to 5 km horizontal resolution with *SMALL* domain. Although the position of the precipitation inversion due to katabatic winds is in agreement with the observed precipitation profile, the amount of simulated precipitation is petite. This is expected since on this type of configuration, the MAR model is very sensitive to ERA-Interim fields outside the simulation domain. Surface precipitation rate is in good agreement with the best MAR configuration with a 5 km horizontal
resolution and a *BIG* domain simulation.

The 25 km horizontal resolution with *BIG* domain represented by green dotted line shows a vertical evolution similar to the standard simulation at high altitude. From 8000 m above the ground level to 2000 m, precipitation accumulation is coherent with the green dashed line simulation but precipitation accumulation still increases when reaching the surface. There does not appear to be any sublimation by katabatic winds. This is also expected, since a grid so deteriorated in resolution does not allow
to correctly simulate small scale processes in MAR.

Figure 3 shows the variation of vertical precipitation accumulation profiles according to two parameters: the horizontal resolution of the grid and the size of the domain in which the MAR simulation is performed. The standard simulation ("5km *BIG*") simulates a correct vertical accumulation profile. However, if the horizontal resolution is deteriorated, MAR no longer simulates evaporation by katabatic winds, and if the simulation domain is reduced, the amount of simulated precipitation is
reduced. These cases are extreme examples of the MAR model, showing that the model remains functional but unusable for configurations too far from its optimal configuration.

Concerning the "25km *SMALL*" simulation, a smaller domain gives less time for the many mesoscale processes to set up in order to stabilize the model and simulate a correct amount of precipitation. Indeed, when we consider a simulation performed on a small domain (250 km x 250 km), the sensitivity of the model to nudging by ERA-Interim reanalysis is high. There
is a transition zone at the domain boundaries from a resolution of 0.75° (∼80 km) in ERA-Interim reanalysis to the 25 km resolution of MAR in the current case. The impact of ERA-Interim reanalysis on synoptic currents of wind, temperature and humidity thus prevents the non-linear processes of the model from being developed.

Despite the configuration of a small domain in the simulation "25km *SMALL*", there is an evaporation of precipitation, which is absent in the simulation "25km *BIG*". We therefore made a comparison between two transects for the particular precipitation



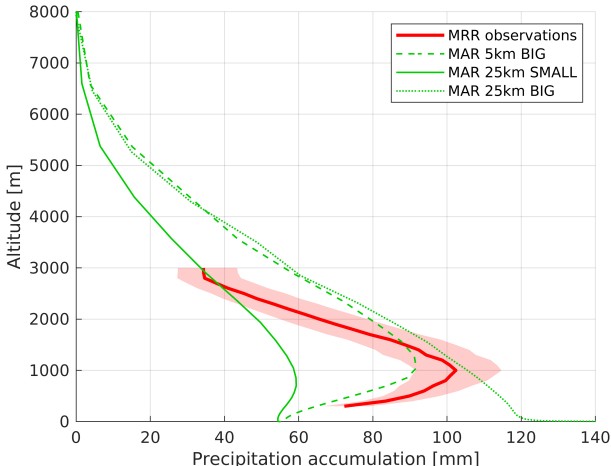

**Figure 3.** Precipitation accumulation profiles of MAR. Green dashed line corresponds to best MAR configuration with a 5 km horizontal resolution and a *BIG* domain. Green dashed line corresponds to analogous MAR simulation to LMDz configuration with 25 km horizontal resolution and a *SMALL* domain. Green dotted line corresponds to the intermediate simulation between to others, with coarse horizontal resolution but *BIG* domain.

event of February 9, 2017 at 6:00 GMT, one crossing the domain of "5km *BIG*" simulation and the other through the domain of "25km *BIG*" simulation. These vertical transects are traced on the Dumont d'Urville (140°E 66.7°S) – Dome C (123.2°E 75°S) axis and are presented in figure 4. In both cases, the katabatic winds are simulated (along the topographical slope). In the "5km *BIG*" precipitation flux decreases sharply near the surface at Dumont d'Urville. There is a dry air pool just above

5   the ocean surface which seems to have a strong sublimating potential. In the "25km *BIG*" this cold air masses could be more horizontally spread so it is not thick enough to sublimate precipitation. Moreover, the precipitation seems to be more diffuse.

### 3.2   Horizontal resolution in LMDz

We have evaluated two horizontal configurations of LMDz with different sizes of the zoomed domain. The *SMALL* configuration is a zoomed domain with a size of 250 x 250 km and the *BIG* configuration is a zoomed domain with a size of 1000 x 1000

10   km. It is important to note that there is the same horizontal resolution inside the zoom. Figure 5 shows the accumulation profiles at Dumont d'Urville resulting from this experiment. The *BIG* simulation simulates a high precipitation accumulation on the surface with 130 mm compared to 55 mm for the *SMALL* simulation. The two simulated precipitation profiles overestimate the observed accumulation profile. The maximum before inversion of the *BIG* simulation is below 1000 m, which is in accordance with the observations. The maximum precipitation of the *SMALL* simulation is at a higher altitude, at 1200 m.





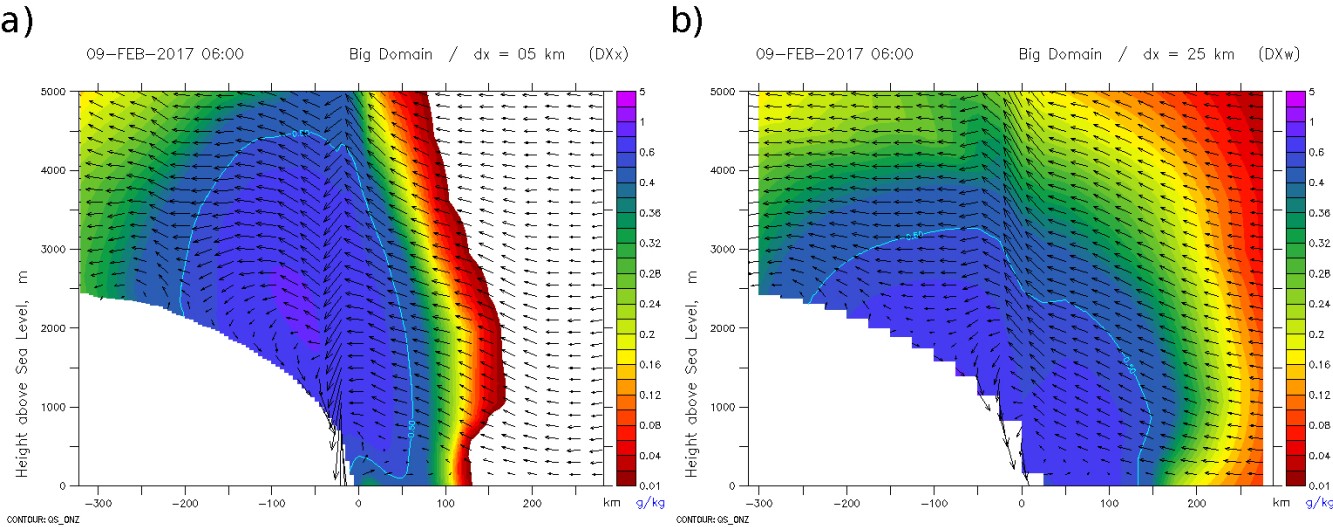

**Figure 4.** Vertical precipitation sections for the particular precipitation event of February 9, 2017 at 6:00 Local Time **a)** crossing the domain of "5km *BIG*" simulation and **b)** crossing the domain of "25km *BIG*" simulation. These cross-sections are traced on the Dumont d'Urville (140°E 66.7°S) – Dome C (123.2°E 75°S) axis. Precipitation is expressed in g/kg. The white contour is 0.5 g/kg..

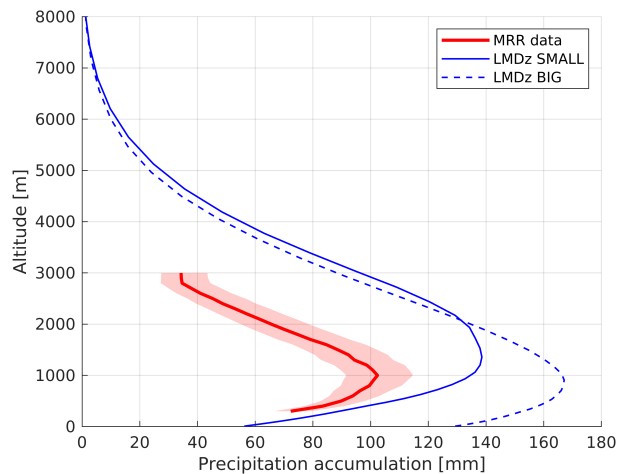

**Figure 5.** Precipitation profiles simulated with LMDz and compared with MRR observations. Blue solid line corresponds to LMDz configuration with a *SMALL* zoomed domain. Blue dashed line correspond to LMDz configuration with a *BIG* zoomed domain. Red solid line is the observed MRR vertical profile of precipitation accumulation and red filled area corresponds to the 95% confidence interval of the MRR observations.



### 3.3 LMDz microphysical parameterizations

Considering that the *SMALL* configuration of the LMDz model is in better agreement with observations than *BIG* configuration (see figure 5), and that the large-scale advected fields are well known thanks to ERA-Interim reanalysis, we performed this experiment in order to evaluate the physics of the model only. Figure 6 presents sensitivity experiments summarized in tables 1

and 2, in comparison with MRR vertical observed precipitation accumulation profile. The surface precipitation rate appears to be in agreement with the MRR at 300 m. However, the amount of simulated precipitation is far too high in all experiments. The maximum precipitation reached by the MRR exceeds 100 mm of accumulation at 1000 m, while the model simulates almost 50 mm more. Moreover, precipitation variations in the simulated profiles, either for the sedimentation rate experiment or the sublimation experiment, are small.

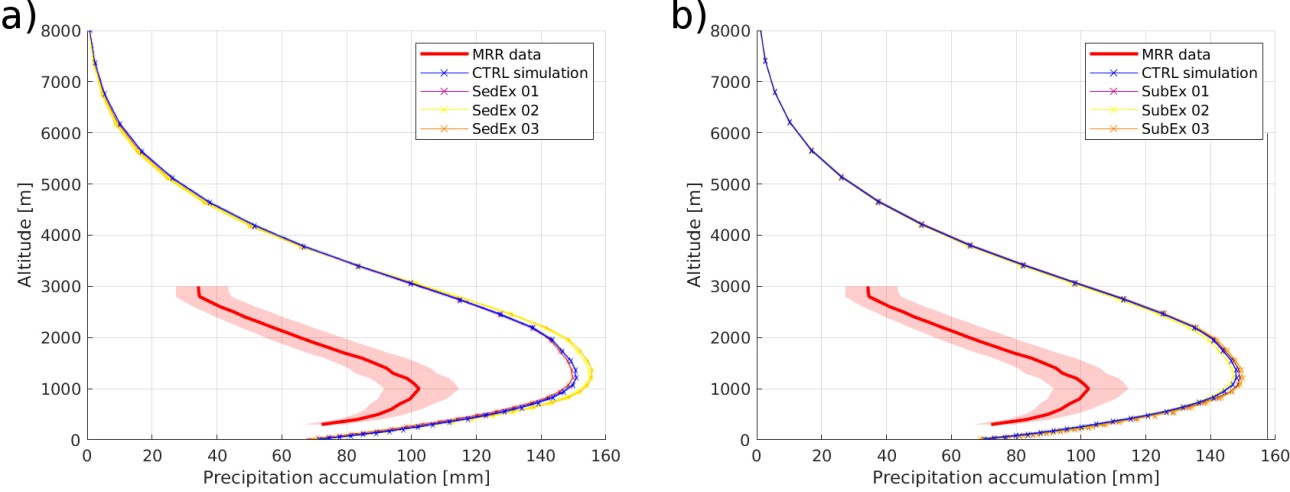

**Figure 6.** Precipitation accumulation profiles of *SMALL* LMDz simulations. Red solid line is the observed MRR vertical profile of precipitation accumulation and red filled area corresponds to the 95% confidence interval of the MRR observations. Blue solid line corresponds to the standard LMDz zoomed configuration with a 25 km horizontal resolution and a *SMALL* domain. Purple, yellow and orange solid lines correspond to sensitivity experiments summarized in table 1 for **a)** and in table 2 for **b)**

### 3.4 Discussion on the resolution and the microphysics

Figure 5 shows a significant difference in the amount of simulated precipitation between *BIG* and *SMALL* LMDz simulations. One of the zoomed regions being small and its circulation very sensitive to ERA-Interim reanalysis while the other being big enough to allow mesoscale circulations to develop without influence from ERA-Interim reanalysis, we verified if the temperature and humidity fields are at the origin of this difference. Figures 7.a and 7.b present the absolute difference in

potential temperature at 950 hPa and 500 hPa respectively between *SMALL* simulation and *BIG* simulation. Figures 7.c and 7.d present the absolute difference in specific humidity at 950 hPa and 500 hPa respectively between *SMALL* simulation and



*BIG* simulation. The maps are at the size of the zoomed region of the *BIG* simulation and the red frames represent the size of the zoomed region of the *SMALL* simulation. For the *SMALL* simulation, the wind, temperature and humidity trends outside the zoomed region are constrained by ERA-Interim. This means that outside the red frame, the *BIG* simulation is closely following ERA-Interim reanalysis. Concerning the temperature, the model in its *BIG* configuration is warmer than the *SMALL*

5 configuration over the continent and colder over the ocean. There is clearly a more humid air mass above Dumont d'Urville in the *BIG* simulation. And in a general way, the continent and the ocean region along the coasts are moister in the *BIG* simulation, with a correlation between temperature and humidity. This shows that mesoscale circulations in the LMDz model redistribute quantities of potential temperature and humidity, thus concentrating moisture along the coasts, as seen on figures 7.c and 7.d, with a warm bias over the Antarctic continent, as seen on figure 7.a and 7.b.

10 Sensitivity tests on the microphysics of LMDz have shown that it has almost no impact on the amount of simulated precipitation. In addition, the amount of simulated precipitation overestimates by approximately 50% the amount of precipitation observed along the vertical profile at Dumont d'Urville. The existing microphysics of the LMDz model does not balance first order warm and moist biases for the representation of polar solid precipitation.



**Figure 7. a)** Difference of potential temperature in LMDz at 950 hPa between *SMALL* and *BIG* simulations. **b)** Same result as **a** at 500 hPa. **c)** Difference of specific humidity at 950 hPa between *SMALL* and *BIG* simulations. **d)** Same result as **c** at 500 hPa. The zoomed area of the *SMALL* domain is represented by the red frame and the zoomed area of the *BIG* domain is represented by the size of the map. The colours range from blue to red. When the *SMALL* configuration overestimates a variable compared to the *BIG* configuration, the color is red.





## 4 Exploring the impact of LMDz numerical dissipation on precipitation

LMDz, like many GCM, contains a dissipation scheme to prevent the accumulation of energy at scales close to the grid reso-
lution. These accumulations of energy appear when GCM is not resolving turbulent scales at the grid resolution (Jablonowski
and Williamson, 2011; Spiga et al., 2018). In the LMDz model, it involves a spatial displacement of dynamic or thermal fields,
which can induce, for example, local warming or a variation in dynamics created by purely numerical processes. Thus, a model
that is too dissipative may generate precipitation that has no physical relevance.

The dissipation is expressed in LMDz as an iterated Laplacian term on a given variable $\psi$ as follows:

$$\left[\frac{d\psi}{dt}\right]_{dissip} = \frac{(-1)^{q_d+1}l_{min}^{2q_d}}{\tau^\psi}\nabla^{2q_d}\psi \tag{4}$$

where $q_d$ is the order of dissipation and $\tau^\psi$ the damping timescale associated with the variable $\psi$ at the smallest spatial scale
$l_{min}$, depending on the horizontal resolution of the model. $q_d$ is an iterative operator, it acts as a filter on the spatial resolution.
When $q_d = 1$, the process is overly dissipative on circulations at large scales and at higher values, dissipation occurs more
at the grid scale than at the large scale. Large values of $\tau^\psi$ means weaker dissipation. Indeed, $\tau^\psi$ represents the time to
dissipate a perturbation on variable $\psi$ developing at the spatial scale $l_{min}$. The three variables designed by $\psi$ are vorticity
and divergence of winds, and potential temperature. They are chosen to set horizontal dissipation on the rotational component
of the dynamic flows ($q_d^{rot}$ and $\tau^{rot}$, i.e. Rossby waves), its divergent component ($q_d^{div}$ and $\tau^{div}$, i.e. gravity waves) and the
diabatic perturbations ($q_d^h$ and $\tau^h$, i.e. latent heat of condensation, rain re-evaporation, snow sublimation, ...).

In LMDz, and more generally in the GCMs methodology, $q_d$ and $\tau^\psi$ are determined empirically. A trade-off between model
stability, damping energy at the smallest scales and minimizing impact on the large-scale flows is sought. There are general
rules for refining the dissipation parameters for LMDz, with $q_d$ ranging between 1 and 4, and $\tau^\psi$ taking values ranging between
one and two hours for a $0.5° - 1°$ GCM simulation. The standard configuration of the LMDz model uses as dissipation values
$q_d^{div} = 1$, $q_d^{rot} = 2$, $q_d^h = 2$ as operators and $\tau^{div} = 600$ s, $\tau^{rot} = 1200$ s, $\tau^h = 1200$ s as timescales.

### 4.1 Sensitivity experiments results

In order to study and understand the impact of the different dissipation parameters on precipitation, we have performed sen-
sitivity tests that are summarized in the table 3. For all sensitivity tests, the resulting simulations are less dissipative than the
control simulation. The corresponding vertical precipitation accumulation profiles are shown in the figure 8. These experiments
were performed on the two configurations of the LMDz under consideration, the results and behaviours are similar but we will
only present those performed on the *SMALL* configuration, which has a precipitation profile closer to the observed profile (see
figure 5).

In a general way, sensitivity experiments on $q_d^{div}$ and $q_d^{rot}$ parameters have little impact on precipitation. The same applies
to the $\tau^{div}$ and $\tau^{rot}$ parameters. However, the dissipation applied to the parameters $q_d^h$ and $\tau^h$ has a strong impact on the
dissipation profile, as observed on the simulations *D03*, *D06* and *D11*. For the *D07* simulation, where all $q_d$ parameters are
modified, it can be deduced that the excellent agreement between the simulated and observed precipitation is due mainly to



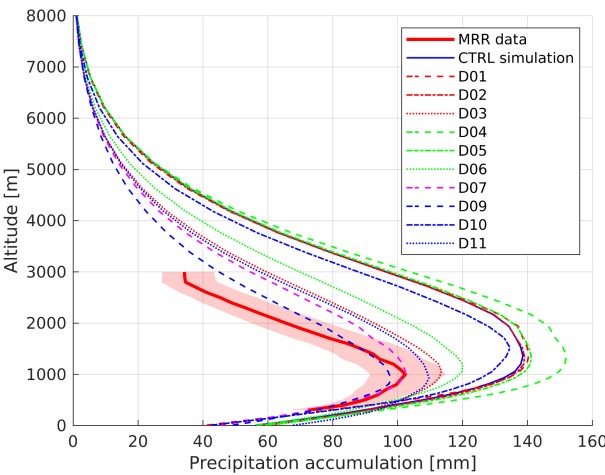

**Figure 8.** Precipitation accumulation profiles of LMDz. Red solid line is the observed MRR vertical profile of precipitation accumulation and red filled area corresponds to the 95% confidence interval of the MRR observations. Blue solid line is the standard LMDz simulation, the red lines represent the experiments on the $q_d$ operators, green lines represent the experiments on the damping timescale $\tau$ parameter. Purple and blue dashed and dotted lines represent experiments on combinations between $q_d$ and $\tau$.

the modifications on diabatic perturbations. Finally, the *D09* experiment best reproduces the MRR observations. Indeed, the simulated profile is very close to the observed profile and within the confidence range of the instrument.

**Table 3.** Dissipation parameter experiments on *SMALL* LMDz precipitation. The values displayed in the table correspond only to tested parameters. When a parameter is not modified, its value corresponds to the standard parameters of LMDz and it is not displayed.

| Experiment | $q_d$ parameter | $\tau$ parameter |
|---|---|---|
| D01 | $q_d^{div} = 2$ | - |
| D02 | $q_d^{rot} = 4$ | - |
| D03 | $q_d^{h} = 4$ | - |
| D04 | - | $\tau^{div} = 1200\text{s}$ |
| D05 | - | $\tau^{rot} = 2400\text{s}$ |
| D06 | - | $\tau^{h} = 2400\text{s}$ |
| D07 | $q_d^{div} = 2 \,;\, q_d^{rot} = 4 \,;\, q_d^{h} = 4$ | - |
| D08 | - | $\tau^{div} = 1200\text{s} \,;\, \tau^{rot} = 2400\text{s} \,;\, \tau^{h} = 2400\text{s}$ |
| D09 | $q_d^{div} = 2 \,;\, q_d^{rot} = 4 \,;\, q_d^{h} = 4$ | $\tau^{div} = 1200\text{s} \,;\, \tau^{rot} = 2400\text{s} \,;\, \tau^{h} = 2400\text{s}$ |
| D10 | $q_d^{div} = 2 \,;\, q_d^{rot} = 4$ | $\tau^{div} = 1200\text{s} \,;\, \tau^{rot} = 2400\text{s}$ |
| D11 | $q_d^{h} = 4$ | $\tau^{h} = 2400\text{s}$ |



## 4.2 Discussion on the dissipation adjustment

In order to study and understand how dissipation affects precipitation, we have investigated the time series of temperatures of the control simulation and the *D09* simulation with the best results relative to the MRR observations. They are presented in the figure 9. The impact of the dissipation is mainly visible at low altitude, where the control model is about 3°C warmer than the *D09* simulation. In addition, when a precipitation event occurs (e.g., February 1, 10, 14, and 21), the control simulation is warmer than the *D09* simulation, which can result in higher precipitation rates being triggered by higher temperature gradients and moister atmospheric masses.

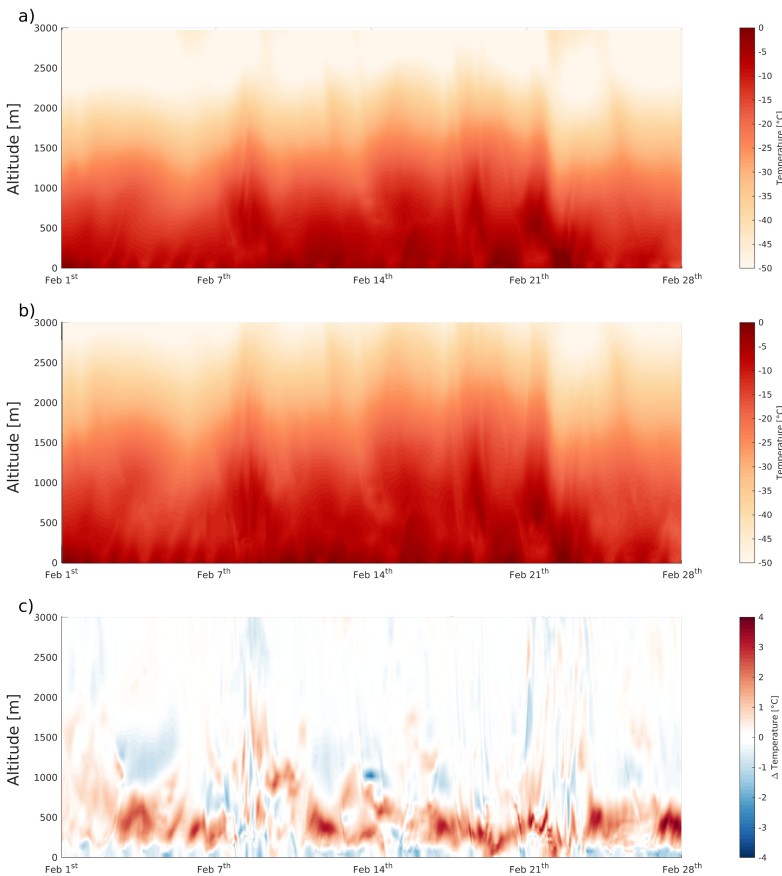

**Figure 9. a)** Control experiment time series of temperature over the February 2017 period. **b)** *D09* experiment time series. **c)** Differential time series of temperature between control and *D09* simulations.

The colors range from blue to red. When the control configuration of the model overestimates temperature compared to the *D09* simulation, the color used is red.

In order to understand the behaviour of the dissipation on a spatial scale, we averaged the temperatures over the month of February according to a transect from Dumont d'Urville (140°E 66.7°S)to Dome C (123.2°E 75°S), showed in figure 10.





When time series are averaged and projected over a larger spatial scale, there is a geographic reorganization of temperature in the less dissipative simulation. In the *D09* simulation, the area above Dumont d'Urville is on average colder than in the control simulation. This is due to warmer temperature fields over ocean regions that are less laterally diffused over Antarctic coastal regions.

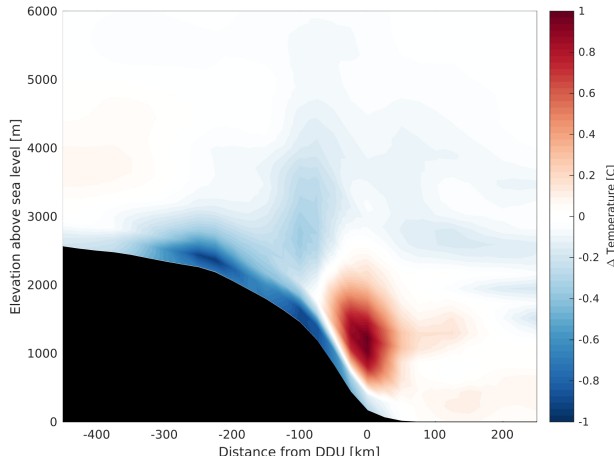

**Figure 10.** Differential averaged temperature between control and *D09* simulations along a Dumont d'Urville (140°E 66.7°S) – Dome C (123.2°E 75°S) Dome C transect.

The colors range from blue to red. When the control configuration of the model overestimates temperature compared to the *D09* simulation, the color used is red.

As shown in figure 11, as the atmosphere cools over the peripheral regions of Antarctica, air masses become less humid and this has a strong impact on precipitation by concentrating it over ocean regions. Thus, the variation in precipitation observed in the figure 8 corresponds to a horizontal redistribution of precipitation in a less dissipative configuration of the LMDz model.

When comparing the MAR model in its optimal configuration with the *D09* simulation of the LMDz model, as shown on figure 12, the average vertical evolution of precipitation is consistent between the two models. This result is interesting because

it shows that a model whose microphysics is simplified to satisfy a global issue can correctly simulate solid precipitation in the Antarctic region. Indeed, the LMDz model only contains a precipitation autoconversion equation and a snowfall resublimation equation, but this allows the climate in Dumont d'Urville to be accurately represented during the month of February 2017, and in particular for the katabatic inversion of precipitation. In the case of the LMDz model, which is too dissipative in its control version, the dissipation adjustment takes priority over the microphysics adjustment and this allows precipitation to be

redistributed over oceanic rather than continental regions. Thus, there is no excess precipitation of purely numerical origin over Dumont d'Urville and having no physical relevance.

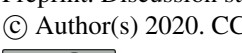



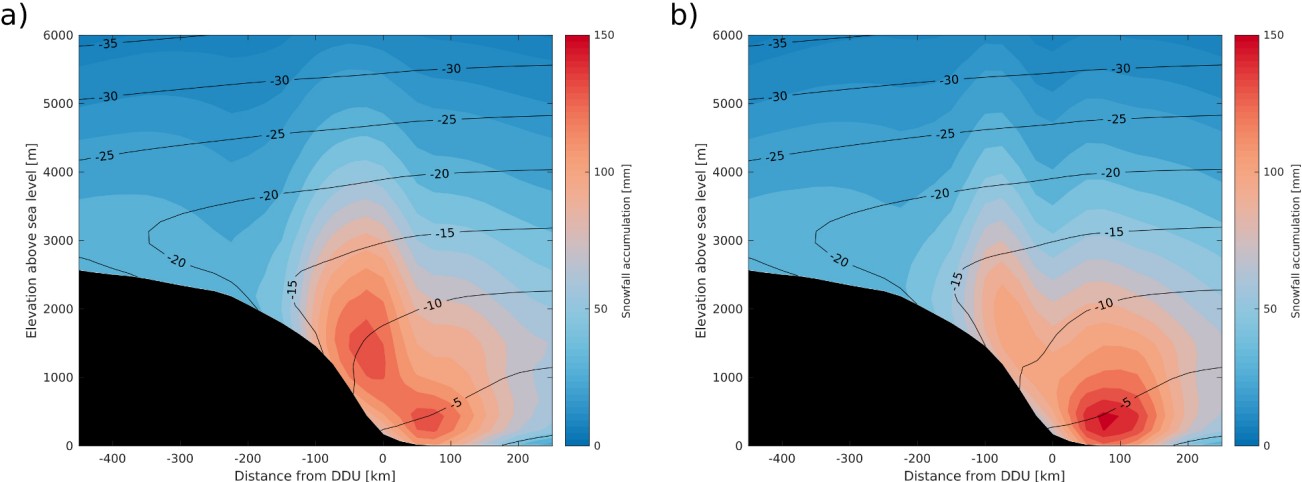

**Figure 11. a)** Average precipitation over the February 2017 period along a Dumont d'Urville (140°E 66.7°S) – Dome C (123.2°E 75°S) transect in the control LMDz simulation. The black lines represent the average isotherms. **b)** Same result for the *D09* LMDz simulation.

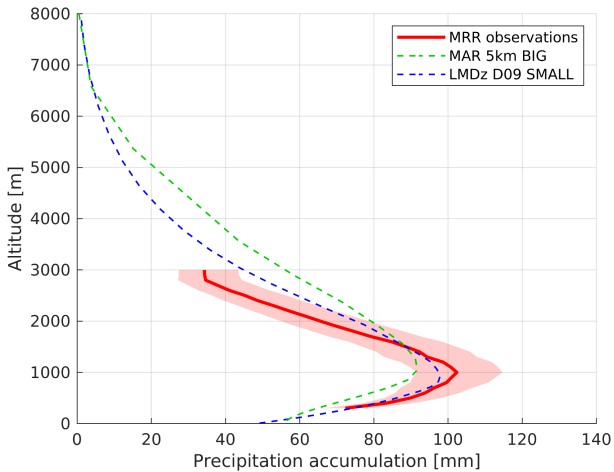

**Figure 12.** Precipitation accumulation profiles of MRR, MAR and LMDz. Red solid line is the observed MRR vertical profile of precipitation accumulation and red filled area corresponds to the 95% confidence interval of the MRR observations. Blue dashed line is the *D09* LMDz simulation, and green dashed line is the *BIG* MAR simulation with a 5 km horizontal resolution.

## 5 Conclusion

Comparison of the vertical precipitation profile observed at Dumont d'Urville with the general circulation model LMDz and the mesoscale model MAR provided a new perspective on precipitation modelling in the polar regions. On the one hand, we evaluated a mesoscale model whose microphysics is complex and which represents accurately the in-situ processes of





precipitation over several horizontal resolutions and configurations. On the other hand, we evaluated a global model in several zoomed configurations over Dumont d'Urville station in order to compare the simulated precipitation profile by testing its microphysics and its numerical dissipation settings with ground radar observations.

The MAR model in its standard configuration (i.e. a 5 km grid resolution and a large domain of 1000 x 1000 km), shown
in the figure 12, accurately replicates the snowfall of February 2017 at Dumont d'Urville. It simulates a 55 mm surface precipitation accumulation with a maximum precipitation of 92 mm at 1100 m, thus showing a correct sublimation of the precipitation by katabatic processes in the lower layers. Above the inversion point of the vertical precipitation accumulation profile, the MAR model seems to simulate too much precipitation. MAR with the same fine grid in a small domain (250 x 250 km) does not simulate enough precipitation because its meso-cyclonic activity is too dependent from the reanalysis fields
to be fully effective. With a coarser grid (25 km of horizontal resolution) in the same domain size, MAR does not efficiently generate the processes related to katabatic winds and dry cold shallow layers above the oceanic surface, which generates too much snow due to an absence of sublimation.

Variations in microphysical parameters related to LMDz precipitation have a small impact on the simulated precipitation profile. However, LMDz is very sensitive to the size of its zoomed region as well as to the advections of large-scale fields
of winds, temperatures and humidity of ERA-Interim reanalysis. Indeed, in a large domain, analogous to MAR standard configuration, where the model is able to generate its own mesoscale circulation, moisture is concentrated above Dumont d'Urville area and warm and moist bias is generated over the continent near the coasts (blue patterns on fig. 7.d). This is not an expected outcome. When a correct general circulation is forced by configuring a small zoomed region where the centre of the zoom remains influenced by the ERA-Interim reanalysis and by improving the GCM dissipation adjustment in a less dissipative
way, the model generates a precipitation profile at Dumont d'Urville that is in excellent agreement with the observed profile.

Numerical parameters that guarantee the stability of a model, such as dissipation, often require empirical adjustments. Dissipation being applied in cases of excess energy to be diffused at the mesh scale, the large-scale currents are not significantly affected by this numerical setting. Thus, the use of observations such as local precipitation rather than large-scale field can be an excellent tool for the fine-tuning of the dissipation of a model, as illustrated here with the LMDz model. This study showed
that a better adjusted GCM model such as LMDz, or a mesoscale model such as MAR are correct for assessing the climate in Antarctica and provide an additional element to the major problem of calculating the mass balance in Antarctica.

## 6 Code and data availability

Data from the Micro Rain Radar at Dumont d'Urville station have been obtained with the logistical support of the French Polar institute IPEV (program CALVA) and are available at https://doi.pangaea.de/10.1594/PANGAEA.882565. The LMDz model is
available from http://web.lmd.jussieu.fr/trac (last access: 9 January 2020). The MAR is freely distributed at http://mar.cnrs.fr/ (last access: 9 January 2020). Due to the size of the high-frequency outputs (several To of simulation outputs) of the LMDz and the MAR models, only simulations of the MAR and the small domain of the LMDz are available: https://doi.pangaea.de/10.1594/PANGAEA.917641.



*Author contributions.* FL led the analysis and drafted the paper. JBM, CC, HG, GK and CG supervised the project. FL ran the LMDz simulations. HG ran the MAR simulations. AC analyzed the simulated precipitation profiles. All authors discussed the results and commented on the paper.

*Competing interests.* The authors declare no conflict of interest.

5 *Acknowledgements.* This work was supported by the French National Research Agency (Grant number : ANR-15-CE01-0003). The authors thank Karine Marquois, Philippe Weil and the IT department of the Laboratoire de Météorologie Dynamique / Institut Pierre Simon Laplace for the informatics support. The authors thank Frédéric Hourdin and Sébastien Froment for their valuable advices.



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
