# Peer review of "Evaluation of coastal Antarctic precipitation in MAR3.9 regional and LMDz6 global atmospheric model with ground-based radar observations"

_The Cryosphere, 2020_

## Referee Comment (RC1) · Anonymous Referee #1 · 8 Sep 2020

Summary

The manuscript evaluates the vertical structure of coastal Antarctic precipitation in the region of Dumont D'Urville station, East Antarctica. The authors compare observations from a micro rain radar with model simulations performed using various configurations of two models: the MAR and LMDz models. Sensitivity tests were conducted to evaluate different model resolutions of MAR and the numerical formulation of processes in LMDz. Adjustments to sublimation and sedimentation in the LMDz model had a minimal effect, whereas dissipation had a large, yet indirect, effect on precipitation.

General comments

This study fits within the aims and scope of The Cryosphere and is of sufficient scientific merit for publication, subject to considerable revision of the text. My main comment is that the text does not currently read like native English, which detracts from the manuscript's quality.

Scientific quality: The authors present a compelling case for the need for improved precipitation modelling in coastal Antarctica and employ relevant scientific methods to evaluate observed and modelled data.

Originality: While the authors do not offer ground-breaking conclusions regarding the scientific understanding of precipitation in coastal East Antarctica, they make a convincing case for adaptations made to optimise the simulation of precipitation with regional and global models, both of which are commonly used in the region because of the scarcity of observational data. For example, MAR is increasingly used to evaluate the surface mass balance in Antarctica – the authors' conclusions show that higher horizontal resolutions and inner domains large enough to resolve meso-scale circulation are essential for accurately simulating precipitation. These are not novel conclusions but add strength to the existing body of knowledge.

Impact: Interestingly, the authors show that numerical dissipation is more important for accurately representing precipitation in East Antarctica using the global model LMDz than physical processes like sublimation and sedimentation, which will be significant for scientists wishing to use this model in the region, but will likely not have broader impact beyond this group.

Presentation: The manuscript is presented in a clear, logical manner with appropriate figures and tables. There are a lot of similar figures (line plots of vertical profiles showing the results of various experiments), and it may be better to think of another way to communicate some of this information. However, my main concern is that the text does not read like native English and can therefore be confusing and distract the reader from

the scientific content. Specifically, this was evident from the order of sentences, verb tenses and non-standard word choices. I have highlighted some instances in my comments but there are too many to comment on individually. I suggest the authors find a native English speaker to proof-read and edit before re-submission.

Specific comments

[Title] The title could be clearer. Some suggestions: "Evaluation of coastal Antarctic precipitation in a global and regional atmospheric model with ground-based radar observations" or "Evaluation of coastal Antarctic precipitation simulated by the MAR3.9 regional model and LMDz global model using ground-based radar observations"

[Figure & Table captions] References to figures and tables should be capitalised throughout. The relevant part of TC manuscript preparation guidelines states: "The abbreviation "Fig." should be used when it appears in running text and should be followed by a number unless it comes at the beginning of a sentence, e.g.: "The results are depicted in Fig. 5. Figure 9 reveals that..."."

[Figures 3, 5, 6, 8 & 12] – these are all extremely similar plots. To keep the reader engaged, is there any other way you could show the differences between models (e.g. scatter plots of observations v model in different height bins, box and whisker plots etc.)? You could possibly combine plots, but you'd have to consider the trade-off between keeping them simple and easy to read (as they are now) and including more information.

[P2, para beginning L10] – Suggest including additional references regarding model simulation of Antarctic precipitation. For example: 1) issues encountered near the coast as a result of the large accumulation gradients (cf. Agosta et al., 2019 – already cited) and steep topography, which models often struggle to represent at sufficient resolution; 2) the role of cloud parameterisations, which are a notoriously stubborn source of model error in the Antarctic and can be important for accurately simulating precipitation (e.g. van Wessem et al., 2018, https://doi.org/ 10.5194/tc-2017-202); 3)

the interplay between the representation of cloud, large-scale circulation and topography, which can produce positive accumulation biases near coasts and negative biases inland (e.g. Lenaerts et al., 2018 https://doi.org/10.1017/aog.2017.42); and 4) model difficulties simulating intense precipitation deposition events like 'atmospheric rivers', which can explain some of the biases over the plateau (e.g. Lenaerts et al. 2018, above).

[P2, L28-29] "...depending on greenhouse gas emissions exercises." – Please insert citation to Palerme et al. (2017), from which I believe these statistics are taken.

[P3, L1-4] Regional models tend to produce more minimal biases. As your study also employs an RCM, could you perhaps include reference to some studies that use higher resolution models over the historical period? E.g. Mottram et al. (2020) https://doi.org/10.5194/tc-2019-333

[P4, L12] Unclear exactly what you mean by "refinement in the boundary layer and troposphere". Please revise.

[P4, 12-13] Without context, it is unclear what you mean by "The vertical precipitation profile studied at Dumont d'Urville in the LMDz model is selected over continental surface." – presumably you mean that model profiles corresponding to the observations are extracted from continental, rather than ocean or ice shelf gridboxes? Please refine for improved clarity. (The same comment also applies to L29-30 regarding the profile selected in MAR)

[P4, L28-29] "MAR is accurate on the surface and in the boundary layer" – do you have a citation to substantiate this? Which parameters are accurately simulated on the surface/in the boundary layer? Further evidence would be helpful to support this claim.

[P5, L15] Explaining your reasoning for focusing on accumulated precipitation instead of a specific event may help the reader.

[Figure 2] Axis and colourbar labels are quite small and difficult to read. Suggest

enlarging the labels (and perhaps labelling only alternate intervals).

[P6, L14-16] How does the representation of topography compare between MAR and LMDz in the SMALL and BIG domains?

[P7, L14-15] "To do this, several orders of magnitude have been fixed to $\beta$ tunable parameter value" – this could be phrased more clearly. For example something like: "Several values of the $\beta$ tunable parameter are chosen that vary across several orders of magnitude"

[Tables 7 & 8] Slightly more detail in the table captions would be helpful. The word ordering is also quite difficult to understand, e.g. "...experiments on LMDz precipitation evaporation" – are these experiments testing the change in precipitation only, evaporation only, or both?

[P8, L6-7] Here is an example of a sentence that would benefit from editing by a native English speaker: "Green dashed line corresponds to best MAR configuration with a 5 km horizontal resolution and a BIG domain is in good agreement with MRR vertical observed profile". This could be revised to (for instance): "The green dashed line shows that the best MAR configuration - with a 5 km horizontal resolution and a BIG domain - is in good agreement with the MRR observed vertical profile"

[P8, L13] The word 'petite' isn't usually used to describe precipitation (although I like the idea of petite precipitation..!) – suggest revising to e.g. 'under-estimated' or 'too small'.

[Figure 3] Please include description of the red line and shaded region (MRR obs + 95% confidence interval) in the caption

[Figure 4] Please include description of what the vectors show in the caption. Again, axis and colourbar labels are quite small and would benefit from being larger. Another small point: as far as I can tell, the 0.5 g kg-1 contour is blue, not white. An inset panel showing the location of the transect may also benefit the reader.

[P12, L4] For clarity, make sure you specifically refer to "potential temperature" rather than temperature.

[P12, L11-12] Another example of where proof-reading by a native English speaker may help with the sentence construction: "In addition, the amount of simulated precipitation overestimates by approximately 50% the amount of precipitation observed along the vertical profile at Dumont d'Urville" could be revised to "In addition, LMDz overestimates the amount of simulated precipitation by approximately 50% throughout the vertical profile at Dumont d'Urville"

[P12, L14] What does this say about the formulation of the microphysics in LMDz? Is it therefore suitable for use in the Antarctic region if changing variables such as sublimation and sedimentation has limited effect? Is further development required to improve the representation of meso-scale processes needed before it can be widely deployed?

[P14, L13] Unclear what you mean "designed by" (do you mean "designated"?) - could you revise your word choice?

[Figure 9] Axis and colourbar labels are again too small to read clearly – please enlarge them. I also think "differential" may not be exactly what you mean here, perhaps revise this part of the captions to "c) Time series of temperature difference between control and D09 simulations" (this also applies to the caption of Figure 10)

[P17, L5] It may aid the reader's understanding to include a brief description of what Fig. 11 shows.

[P17, L9-16] This is indeed an interesting result! Are you able to speculate about why this might be?

[Figure 12] Nice summary figure. You could sign-post the reader to this figure more to emphasise your most important take-home results, for instance by including a brief sentence summarising what it shows.

[P19, L23-25] Did you examine the effect of tuning the dissipation variables on any other fields? Does tuning the model to better represent one or two variables in one geographical area have knock-on effects on other variables, for instance introducing competing biases and errors elsewhere in the model? Can you conclusively say that amending the representation of dissipation does not produce other cancelling errors (i.e. that the model gets the 'right results for the wrong reasons')?

---

## Referee Comment (RC2) · Anonymous Referee #2 · 30 Sep 2020

General Comments

This manuscript examines the vertical profile of precipitation over Dumont d'Urville research station in Antartica, comparing surface based radar observations to two models: the meso-scale MAR model, and the general circulation LMDz model. Sensitivity tests are conducted with the MAR and LMDz models to explore the changes to snowfall representation that result from modified domain size, resolution, and various parameterizations.

[Figure]

The introduction talks broadly about precipitation over Antarctica in models and observations, but the focus of the analysis is on a particular coastal location. It would be helpful for the introduction to more clearly set up the importance of the coastal analysis. The methods section needs to be more precise regarding the experimental setup. Because the methods are slightly vague, it is difficult to interpret some of the results (see specific comments below). The detailed sensitivity experiments will likely be of great interest to cryosphere researchers within the LMDz and MAR modeling communities, and perhaps of general interest to the larger precipitation modeling community. The conclusions are nicely presented, highlighting the interesting result that dissipation plays a relatively more important role than microphysical parameterizations in LMDz precipitation over Dumont d'Urville. Overall, this manuscript would benefit from a detailed edit of the text for clarity and flow. I have pointed out some specific instances where the text should be improved, however there are too many to call out each individually. The wording/language issues detract from the analysis and make it hard to follow the narrative.

Specific Comments

[P2L6] The first sentence requires a citation.

[P2L10] While precipitation is the largest positive contribution to the surface mass balance, perhaps it would be worth acknowledging water vapor deposition here as well.

[P3L1] "And even though the simulated surface precipitation is compared to an observation level at an altitude of 1200 meters above the local surface, the discrepancy between data and models is large, and questionable for the future prediction of precipitation." This sentence implies that the satellite rate is not appropriate to compare to the surface. The CloudSat data product used in Palerme et al., 2017, 2C-SNOW-PROFILE, provides a surface snowfall rate estimate based on the reflectivites aloft. Consider rephrasing.

[P3L3] "In addition, the agreement between data and models is even worse for the

simulation of precipitation on the plateau than over the peripheral regions (Palerme et al., 2017; Roussel et al., 2019)." Since the manuscript analysis is focused on a coastal location, this does not seem relevant for the introduction.

[P4L4-10] This part of the methods section is a bit unclear. Are the "surface schemes" that provide forcing of the atmosphere from below based on climatology? Perhaps it would be helpful to include a map showing the location of the station as well as the region that has been stretched. At a minimum it is important to include the latitude and longitude of the study area, giving an indication of how far away the "nudging" is occurring.

[P4L12] "The vertical precipitation profile studied at Dumont d'Urville in the LMDz model is selected over continental surface." A map would be helpful to understand what you say here. How large is the region used for the vertical precipitation profile? Do you mask out information over the ocean?

[Figure 1] I am not familiar with looking at vertical profiles of accumulation. Since snow does not actually accumulate above the surface, perhaps a vertical profile of rates would make more sense?

[P6L11] "allows these simulations to be run as "critical" cases of MAR use" For a non-MAR specialist, it is unclear what this means. This sentence does not seem necessary for understanding the modifications you've made.

[P7L9] "The different imposed values are summarized in table 1" Why were these particular values chosen?

[P8L3 and Figure 3] Are these vertical profiles only the values over the continental surface (as mentioned in the methods), and if so, which grid boxes are averaged within the BIG and SMALL domains to produce the profiles. Perhaps the grid boxes used could be drawn in Figure 2?

[P8L30] "We therefore made a comparison. . ." I am unclear of the connection between

the preceding sentence and the comparison this statement goes on to describe. Consider revising to make the connection clearer.

[Figure 4] An x-axis label is needed. The location of Dumont d'Urville is referenced in the text, but it is unclear where the research station is in the figure.

[P9L7] "3.2 Horizontal resolution in LMDz" In the text of this section, you are clear that the horizontal resolution is the same inside the zoom, so perhaps a section heading like "Horizontal configuration in LMDz" would be more appropriate?

[P12L3] "This means that outside the red frame, the BIG simulation is closely following ERA-Interim reanalysis." Looking at your Figure 7, it seems that the area outside the frame is showing differences in all four panels. It would be helpful to have context as to what difference values would be considered large for the variables.

[Figure 7] What is the reasoning for presenting the differences at these two particular levels (950 and 500 hPa)? In the text, I do not see a discussion of one level vs the other, so it is unclear why both are included.

[P16L5] "when a precipitation event occurs (e.g., February 1, 10, 14, and 21)" Perhaps highlight precipitation time periods in Figure 9 so they are more clear to the reader? Do the control and D09 simulations always precipitate at the same time intervals?

[P16L6] "triggered by higher temperature gradients and moister atmospheric masses" It is not clear from Figure 9 that there are stronger temperature gradients or moister air masses. Perhaps reference supporting figures/references/analysis that can support this statement?

[Figure 11] This figure highlights how strong the gradient in precipitation is near Dumont d'Urville. With such a sharp gradient, it seems very important to document what region (location and size) is averaged to get your model vertical precipitation profiles. For example, in Figure 12, where the MAR 5km resolution and LMDz 25km resolution vertical profiles are compared, how many/which grid boxes are averaged? Are multiple

MAR grid boxes averaged to cover the same spatial region as LMDz? Recommend clarifying this in the methods section.

[P18L18-20] "When a correct general circulation is forced by configuring a small zoomed region where the centre of the zoom remains influenced by the ERA-Interim reanalysis and by improving the GCM dissipation adjustment in a less dissipative way, the model generates a precipitation profile at Dumont d'Urville that is in excellent agreement with the observed profile." Is the suggestion here that the dissipation adjustment could/should be made to the standard setup of LMDz for more accurate snowfall representation in coastal Antarctica? Is it likely that precipitation characteristics in other regions would be improved/changed by such an adjustment?

Technical Corrections

[P1L6] MAR isn't defined

[P2L8] remove period in "mm.yr-1"

[P2L24] "The calculation time is crucial", crucial to what? Consider rephrasing this sentence for clarity.

[P4L1] "water vapor" instead of "vapor water"

[P4L20] "according to (Lin et al., 1983)" remove the parentheses.

[Figure 8] Recommend changing the line color for better contrast. Particularly for D07, since it so closely resembles D01.

[Figure 9] Remove the hard return between the end of the first sentence and the start of the second in the figure caption.

---

## Referee Comment (RC3) · Anonymous Referee #3 · 5 Oct 2020

This is a weak paper that should be rejected as it will take a huge amount of work to rectify.

The key aspect to the precipitation profile in Figure 1 is the sublimation of the falling precipitation by the dry katabatic winds as described Grazioli et al. (2017b). I would have expected the output from MAR3.9 in the immediate vicinity of Dumont D'Urville to be fully exploited to examine the precipitation generation, the sublimation and their causes, namely the winds along and normal to the DD-Dome C transect, the temperature field, the relative humidity field, the sublimation profile, the vertical motion field,

along with the precipitation profile. Then the differences with the control run could be explained in terms of these causes. I still don't know why, in physical terms, there is the difference between the 5 km and 25 km domains, and what role(s) the simulated katabatic winds play. Is there pooling or blocking of the airflow? Where does the dry air pool come from in 5km BIG (Fig. 4, barely visible, where is the white contour)? Why focus on the monthly precipitation accumulation? What about the individual precipitation events? Etc.

Then there is the tuning exercise for LMDz6 for February 2017. The real test for the validity of this tuned precipitation prediction is to try it out for a month independent of the tuning.

Figure 8 is very challenging to discriminate between all the different lines, dashed, dotted, and solid, many of which are very similar.

---

## Short Comment (SC1) · 6 Oct 2020

Dear authors,

As the main developers of the regional climate model MAR, we would like to raise strong reservations on the work proposed with MAR in this paper because of inconsistency of the methodology followed to analyze the sensitivity to horizontal grid size. Our comments also include recommendations for the description of the model, very diverging in its current state from what we do try to follow in our respective publications.

[Figure]

Although LMDz has been mostly designed for representing the atmospheric dynamics over large integration domains over which the model is much more free to evolve, the evaluation of the model ability to represent over small domains observed precipitation profiles at DDU and the sensitivity analysis to resolution and dissipation are very interesting. Regarding MAR, the description of the model must be more thorough, and simulations at 25 km resolution over the SMALL (10x10 pixels) and BIG (40x40 pixels) domains cannot be mentioned due to the too weak number of pixels that prevents MAR to simulate its own circulation, making the results meaningless. We strongly recommend to discard these results from the paper and to focus more on the LMDz simulation and possibly on the 5 km simulation with MAR to strengthen the quality and the scientific robustness of the paper. Our comments are reported below.

Cécile Agosta, Xavier Fettweis, Christoph Kittel and Charles Amory

1. Please double check which version of MAR was actually used in the study since only MAR4 is able to go down 15 cm in vertical resolution. Moreover, any version of MAR3 does not enable simulation on a domain smaller than 60x60 grid points, as done in this study. Any currently published study with MAR3 below version 3.11 does not include a blowing-snow scheme, which has only been evaluated and made functional recently. If the study has been carried out with MAR4, it should appear clearly in the text (and the title) without ambiguity to keep consistency from one evaluation exercise to another and prevent confusion between the actual capabilities of different model versions. In that case, referring also to the performance of MAR3 (stable version) to support the use of MAR4 (currently in development) must be avoided given the too large differences between the two models. If the version is truly MAR3.9, this must be stated as is in the text but the description of the model set-up needs to be adapted to fit with Kittel et al. (2018) and Agosta et al. (2019). Please adapt the code and data availability section accordingly since MAR4 has not been made publicly available yet.

2. P3 L24: The atmospheric radiative transfer in MAR3/MAR4 is adapted from Morcette (2002).
3. P4 L20-24 : Avoid wording "accurately" when it's not supported by good agreement with observations. MAR3 has only been "officially" extensively evaluated so far against near-surface meteorological observations, and against few atmospheric profiles collected in stable environments very different from your measurement location for which local turbulence schemes (as implemented in MAR) are known to misrepresent the well-mixed character of neutral ABL, and thus U, T, RH and ultimately sublimation profiles. From this perspective, we suggest to reformulate with "MAR includes a detailed representation of the ABL" and refer to some relevant publications. Similarly, the blowing-snow scheme is not activated nor evaluated in Agosta et al. (2019), neither the interactions with katabatic winds. Please reformulate this as well. Sublimation of blowing snow may influence the relative humidity and then the sublimation potential of precipitation within the lowest part of the atmospheric boundary layer (Grazioli et al., 2017b), which falls well within the scope of the study. Finally, note that the very small additional computational cost required for activation of the blowing-scheme cannot be reasonably used as a justification for disabling it. If you're using MARv3.9, use the same justification than in Kittel et al. (2018) or Agosta et al. (2019).

4. P4 L25: + surface pressure at the model lateral boundaries + SST and SIC at the surface of the ocean.

5. P4 L26: "optimal" may be too emphatic. There is no comparison nor sensitivity analysis that would support this configuration is more advisable than other ones likely to lead to similar results.

6. P4 L28-29: The fine discretization close to the surface is not a sufficient argument to state that MAR is accurate in the absence of quantitative comparison with observations. Please change wording.

7. P4 L30: How has the duration of 4 months been chosen for the spin-up of MAR? It seems that the authors refer here to the spin-up needed for the atmosphere to reach equilibrium, not for the snowpack (which would require a longer spin-up). Is there any

reason justifying why MAR would require a shorter spin-up than LMDZ?

8. P6 Section 2.4 + P8 section 3.1: Contrary to LMDz, MAR is only forced at its lateral boundaries by surface pressure, temperature, specific humidity, and wind speed. The formation of clouds and resulting precipitation can therefore only be achieved after a considerable distance away from the boundaries, and so it is for the fine-scale circulation. For any simulation, the area of interest must be located at least 20 pixels away from each of the 4 lateral boundaries. Running a RCM over a domain that actually corresponds to the relaxation zone (SMALL 25km, 10x10 pixels) is meaningless. This is more or less an interpolation of the ERA-Interim large-scale fields on the model grid without any cloud nor precipitation, and does not constitute any model result. Similarly the size of the BIG domain is also prohibitive for the run at 25 km resolution because MAR is still too influenced by the large-scale forcing at the center of its domain. Without enlarging the domain at 25 km, these results (and every related comment) could not be used for a reasonable scientific demonstration and must be removed from the study. Note that using MARv3, Franco et al., (2012) showed that precipitation remains globally unaffected by changes in resolution and even obtained inverse results from that presented here (the higher the resolution, the larger the precipitation rates).

9. Fig. 2 : Indicate DDU on the MAR continental pixel from which the modelled vertical profile has been extracted.

10. P6 L10-12: The reference to Franco et al. (2012) is inappropriate here since the resolution they used reflects the compromise between computational time and model accuracy due to the large number of simulations, but not model capabilities. Please remove.

11. P8 L25-26: As explained above, this is a matter of scientific meaning and not model capabilities of former MAR versions or "not optimal" configurations. Please remove.

12. P8 L9 : Consider linking this bias not to MAR but to the version of the model used in this study since biases can change depending on the version. This is why it is of first

importance to specify the version used.

Agosta, C., Amory, C., Kittel, C., Orsi, A., Favier, V., Gallée, H., van den Broeke, M. R., Lenaerts, J., vanWessem, J. M., van de Berg,W. J., et al.: Estimation of the Antarctic surface mass balance using the regional climate model MAR (1979-2015) and identification of dominant processes, Cryosphere, 13, 281–296, 2019.

Franco, B., Fettweis, X., Lang, C., and Erpicum, M.: Impact of spatial resolution on the modelling of the Greenland ice sheet surface mass balance between 1990–2010, using the regional climate model MAR, The Cryosphere, 6, 695–711, https://doi.org/10.5194/tc-6-695-2012, 2012.

Grazioli, J., Madeleine, J.-B., Gallée, H., Forbes, R. M., Genthon, C., Krinner, G., and Berne, A.: Katabatic winds diminish precipitation contribution to the Antarctic ice mass balance, Proceedings of the National Academy of Sciences, 114, 10 858–10 863, 2017b.

Kittel, C., Amory, C., Agosta, C., Delhasse, A., Doutreloup, S., Huot, P.-V., Wyard, C., Fichefet, T., and Fettweis, X.: Sensitivity of the current Antarctic surface mass balance to sea surface conditions using MAR, The Cryosphere, 12, 3827–3839, https://doi.org/10.5194/tc-12-3827-2018, 2018.

Morcrette, J.-J.: Assessment of the ECMWF model cloudiness and surface radiation fields at the ARM SGP site, Mon.Weather Rev., 130, 257–277, https://doi.org/10.1175/1520- 0493(2002)130<0257:AOTEMC>2.0.CO;2, 2002.